# Effect of External Charging on Nanoparticle Formation in a Flame

**DOI:** 10.3390/ma14112891

**Published:** 2021-05-28

**Authors:** Elena Fomenko, Igor Altman, Igor E. Agranovski

**Affiliations:** 1School of Engineering and Built Environment, Griffith University, Nathan 4111, Australia; elena.fomenko@griffithuni.edu.au; 2Combustion Sciences and Propulsion Research Branch, Naval Air Warfare Center Weapons Division, 1 Administration Circle, China Lake, CA 93555, USA; igor.altman@navy.mil

**Keywords:** MgO nanoparticle, combustion synthesis, external particle charging

## Abstract

This paper attempts to demonstrate the importance of the nanoparticle charge in the synthesis flame, for the mechanism of their evolution during formation processes. An investigation was made of MgO nanoparticles formed during combustion of magnesium particles. The cubic shape of nanoparticles in an unaffected flame allows for direct interpretation of results on the external flame charging, using a continuous unipolar emission of ions. It was found that the emission of negative ions applied to the flame strongly affects the nanoparticle shape, while the positive ions do not lead to any noticeable change. The demonstrated effect emphasizes the need to take into account all of the phenomena responsible for the particle charge when modeling the nanoparticle formation in flames.

## 1. Introduction

The advances made in the nanotechnology field over the past years are due to the involvement of synthesis method and classification facilities from physics, chemistry and biology fields, and are combined to produce new materials with enriched properties. In particular, magnesium oxide (MgO) nanoparticles represent one of the promising classes of nanomaterials due to their biocompatibility, lightweight property, rapid metabolic activity and, therefore, encouraging applications of the formed particles in diverse disciplines [1].

The prospects for applying MgO in various sectors including biological [1], electronics [2], coating [3] and synthesis of petrochemical products [4] comprise the foundation of the advances in aerosol technology, including synthesis of nanoparticles through direct metal combustion methods. The literature data highlighting nanoparticle synthesis through combustion techniques presents an increased interest in present times due to potentials to accomplish high purity and controlled product properties. Nevertheless, the dispersed nature of MgO, controllable fabrication of nanoparticles with various surface morphologies, and the understanding of growth mechanisms are still a challenge. Notably, due to the high surface-to-volume ratio at the miniature scale, nanostructure morphology has much influence on their properties.

Understanding the mechanism of nanoparticle formation from vapor is essential for development of gas-phase nanotechnology. It is of a particular interest for flame synthesis, the commercially used method for nano-oxide production [5]. In addition, nanoparticles formed by combustion possess unique properties that distinguish them from counterparts synthesized by other methods, such as solgel [6]. The consistent description of a particle size evolution in the system is needed in order to tune the final product for specific applications. However, there is no consensus on how the particle formation occurs in flames. The common approach is to model the particle growth using the aerosol concept, i.e., considering nucleation/coagulation/coalescence as the major processes that govern the transition from vapor to condensed particles [7]. It has been demonstrated that this approach does not to work in the case of MgO nanoparticle formation during Mg combustion in the air [8]. Being of a cubic shape in the synthesis flame, these solid particles could not be generated via steps that would occur with liquid particles. This has been used as an argument for why MgO nanoparticles are dominantly formed via condensation (surface growth). At the same time, it is not clear whether coagulation/coalescence may occur in different systems with different materials synthesized, where particles are molten, so the solid state of particles does not preserve them as would be in the case of MgO. Addressing this issue seems to be important in order to understand whether or not condensation is the general process which controls the nanoparticle formation in the system of interest. This first requires analyzing whether the solid state is the actual reason why MgO particles are not being coalesced in the experiment, or if it was just a lucky coincidence that allowed for eliminating the coalescence-based explanation of the particle growth while interpreting the observed results. 

In the current paper, we suggest that the main cause of the absence of MgO particle coagulation is the suppression of particle collisions due to an inherent particle charging. The latter occurs due to the thermionic emission of electrons, so the nanoparticles are positively charged. The Coulomb repulsion between the particles hinders collisions. Unlike a particular case of formation of solid MgO particles, the particle charging due to the thermionic emission of electrons, which leads to the collision suppression, looks general for the flame-generated particulate because the particles are always synthesized in flames at a high temperature. Then, a demonstration of the particle charging effect on the particle evolution in the synthesis flame, which is the subject of the current communication, is an important step to verify the mechanism of the particle growth. In order to consistently interpret the results of charging, studying the MgO nanoparticles remains the best way due to the cubic particle shape.

## 2. Experimental

This experiments program was conducted as described in our previous work [8,9,10]. The approach is sketched in Figure 1. In brief, a magnesium cube with ~3-mm side with a blind hole with a 0.5-mm diameter was secured on a 0.25-mm diameter tungsten wire, and ignited by a portable propane–air diffusion flame that was immediately removed after the particle ignition. The typical burn time was on the order of 10 s. A tweezers-type support holding the transmission electron microscope (TEM) grid was attached to a vertical rod. The quick rod rotation around the vertical axis allowed for the very short residence time of the TEM grid within the zone of MgO nanoparticle generation. Some estimates indicated that the residence time of the grid in the generation zone was about 1 ms in all experiments [8].

We collected MgO nanoparticles in the unaffected flame and in the flame to which the unipolar emission of ions was applied. Both the negative and positive ion emissions were utilized and generated by a powerful home-made ionizer capable of producing both positive and negative ions. The ionizer output was tested by the Scanning Mobility Particle Sizer Spectrometer (DMA Model: 3080N, and CPC Model: 3775, TSI Inc., Shoreview, MN, USA) to ensure that no particulate was generated, so the flame products would not be contaminated. The MgO nanoparticles deposited on the grids were analyzed with TEM (JEM-1400, JEOL Ltd., Tokyo, Japan). The comparison of TEM images of particles collected in the unaffected and charged flames was performed.

## 3. Results and Discussion

The typical TEM image of nanoparticles collected from the unaffected flame is presented in Figure 2a, along with that of the particulate collected from the flame exposed to positive ions (Figure 2b). As it is seen, the particles from both flames look similar. We were unable to find any significant difference between nanoparticles produced in the unaffected flame and the flame to which positive ions were applied. Both flames produce nanoparticles of a cubic shape, with the clearly distinguished sharp edges of cubes. This nanoparticle shape and size, mainly ranging between 20 nm and 200 nm, is consistent with that reported in our previous work [8].

On the contrary, the exposure of the flame to the negative ions led to a substantial change of the nanoparticle shape. The TEM images of nanoparticles collected from the corresponding flame are shown in Figure 3. As is clearly seen, many nanoparticles have lost ideal cubic shape; their edges are no longer so sharp, and some of the nanoparticles tend to be spherical. 

Such a noticeable change of the nanoparticle shape in the case of the flame exposure to the flux of negative ions could only be explained by the fact that the nanoparticle temperature in the corresponding flame is substantially higher than the temperature in the unaffected flame and the flame to which positive ions were applied. The elevated temperature, which is presumably close to the magnesia melting point, provides conditions for the partial melting of nanoparticles. This leads to the edge smoothening and the spheroidization of cubic particles. It should be noted that this concept is in line with the temperature estimate of ~2600 K, corresponding to an Mg particle combustion [11] that is well below the magnesia melting point of about 3100 K. Since both growing (hot) and grown (cold) particles coexist in the flame [12,13], the pyrometry temperature measurements in the system of question would not be representative. In addition, it must be emphasized that, regardless of the size, no particles with a rounded shape are seen in both scenarios: unaffected flame and that exposed to positive ions. In contrast, rounding of edges is observed in the case of negative ions exposure for the entire range of aerosol sizes. Such observation suggests that melting-point depression is a very unlikely scenario in the proposed nanoparticle generation process.

The overheating of the nanoparticles in the flame exposed to negative ions can be possible only if heat transfer of these particles with the environment is less intensive compared to the nanoparticles in the unaffected flame and the flame to which positive ions are applied. The proposed mechanism for suppressing heat transfer is as follows. MgO nanoparticles formed in the flame of a burning Mg particle are positively charged due to thermionic emission of electrons [10,14]. It is natural to assume that the presence of the negative ions in the system significantly suppresses the thermionic emission of electrons with the same electric charge sign. At the same time, it is very unlikely that positive ions could affect this emission. Thus, thermionic electron emission is a phenomenon that can control heat transfer between a nanoparticle and the environment. 

As far as we know, this type of heat transfer, i.e., associated with thermionic electron emission, has not been discussed in the literature. The reason is that the proposed effect can only take place in a non-isothermal system, where the temperature of the nanoparticle differs from the temperature of the gas bath. Note that the existence of this non-isothermality was predicted in ref. [12], and has recently been demonstrated in refs. [13,15]. The reason of this non-isothermality is an interplay of different mechanisms of heat transfer between the nanoparticle and surrounding environment. While the thermal radiation plays the major role, conductive heat transfer sustains the steady-state particle temperature [12,15]. 

The temperature difference between the nanoparticle and gas leads to the appearance of an energy flux on the particle interface, which is related to the thermionic emission. Under steady-state conditions, i.e., for an already charged nanoparticle, the number of electrons leaving the particle due to the thermionic emission is equal to the number of electrons captured by the particle from the surroundings. The electrons leaving the particle and those captured by the particle have different temperatures due to the particle–gas non-isothermality, which results in the effective energy flux, i.e., heat transfer. Then, the efficiency of the heat transfer of interest depends on the flux of electrons leaving the particle due to the thermionic emission under steady-state conditions. This flux depends on the nanoparticle surroundings, in particular on the nanoparticle volume concentration, temperature, etc. The consistent quantification of this flux is required in order to describe the effect in detail. Here, we will try to demonstrate that the electron flux is sufficiently high in order to justify the significance of heat transfer associated with thermionic emission.

The flux of electrons leaving the particle due to the thermionic emission can be expressed as [16],
(1)$$J_{e}=k\frac{A}{e}T^{2}\exp\left(-\frac{W}{k_{B}T}\right)\;,$$
where the Richardson constant for thermionic emission *A* = 1.2 × 10^6^ A m^−2^ K^−2^, *e* is the elementary charge, *k* is the material related constant, which is on the order of 0.3–1, *k_B_* is the Boltzmann constant, *T* is the particle temperature, and *W* is the material work function.

The magnesium oxide work function is about 4.5 eV [17]. Then, from Equation (1) at *T* = 3000 K, one can obtain the flux of electrons, *J_e_*, on the order of ~10^24^ s^−1^. This flux is significantly lower than the flux of gas molecules colliding with the particle surface. This gas molecule flux can be expressed as,
(2)$$J_{g}=\frac{n_{g}v_{g}}{4}\;,$$
where *n_g_* is the gas molecule concentration and *v_g_* is the average gas molecule velocity. At the flame conditions it can be estimated as *J_g_*~10^27^ s^−1^. 

Unlike the gas molecules, which have an energy accommodation coefficient of about 10^−3^ due to their “surface” interaction [18,19], the efficiency of the electron energy transfer in the suggested process can be equal to unity. Then, despite the fact that the electron flux is a couple of orders of magnitude lower than the flux of gas molecules, it can provide a comparable heat flux. Therefore, any noticeable influence of electrons on the thermionic emission leads to a change in the temperature regime of the nanoparticle, which is reflected in the partial melting of particles in a flame exposed to the negative ions. 

In order to quantify the suggested mechanism of the electron-related heat transfer, a concentration of the free electrons around the nanoparticle could be evaluated to provide steady-state conditions of the process. Under these conditions, the number of the emitted electrons must be equal to the number of the captured electrons. The latter one is given by the same expression as Equation (2) with the electron-related values *J_e_*, *n_e_* and *v_e_* entering the equation. At *J_e_* = 10^24^ s^−1^, one can achieve *n_e_*~10^18^ m^−3^ = 10^12^ cm^−3^. 

The ion concentration measurements were then undertaken by an Air Ion Counter (AlphaLab Inc., Salt Lake City, UT, USA), commonly used for similar investigations [20]. Similarly to the previously reported results, measurements undertaken in a close vicinity of the ion emitter exceeded the upper limit of the instrument (2 × 10^6^ cm^−3^). This outcome was well expected for such powerful emitters, dictating a need for some extrapolation of the results obtained at more distant locations from the device. The results of measurements and corresponding analysis are shown in Figure 4. Obviously, taking into account a substantial increase in the ion concentration over very short distances, Figure 4 should rather be considered as a generalized illustration of the ionizer capabilities. As is seen, the upper detection limit of the instrument was reached at the distance of approximately 10 cm from the emitter. Then, the results gradually decreased over the following 30 cm, reaching low values comparable with the natural background. The extrapolation of the graph shows that the concentration of ions at the source can be potentially on the order of 10^12^ cm^−3^, i.e., be comparable with the estimated electron concentrations. Such findings enable one to draw the corresponding conclusion that the ion flux applied to the flame can significantly affect the charge distribution in the system, and, therefore, result in the particle temperature change. A detailed study of the charge distribution would be helpful, however, due to a limited process time, the existing technique such as that described in refs. [21,22] should be substantially modified. 

It must be noticed that, although the nanoparticles in the flame exposed to negative ions look partially molten, there is no noticeable agglomeration; the particles are rather individual. It additionally confirms the occurrence of the particle Coulomb repulsion and rules out the coagulation/coalescence route for the particle growth.

## 4. Concluding Remarks

The reported change of the MgO nanoparticle shape depending on the flame exposure to ions justifies the importance of charging phenomena for the particle formation in flames. For charged particles, condensation is the major process responsible for the particle growth. The thermionic emission, which controls the inherent particle charging, should be taken into account for the consistent description of the evolution of nanoparticles generated in the high-temperature flames. It is likely a process that may significantly affect the heat transfer on the nanoparticle interface with the bath gas. 

The shape change we observed in the current work under certain charging conditions is the clear indication of the effect we found. Although we provide a qualitative rather than a complete direct explanation of this effect, our findings prove the presence of the particle charge in the unaffected flame, and, correspondingly, an essential role of this charge in the nanoparticle formation.

## Figures and Tables

**Figure 1 materials-14-02891-f001:**
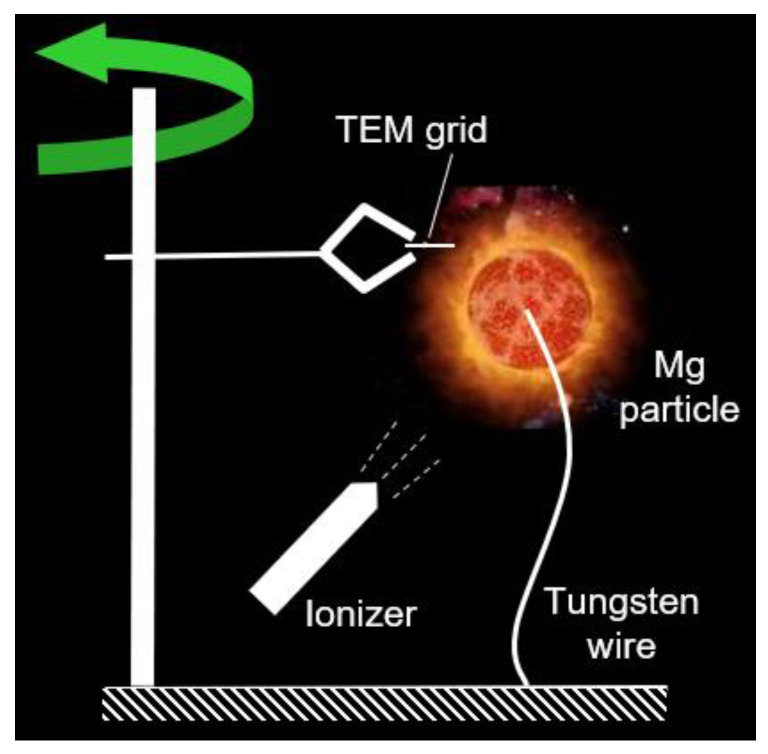
The sketch of the experimental set-up utilized for collection of MgO nanoparticles from a single Mg particle flame.

**Figure 2 materials-14-02891-f002:**
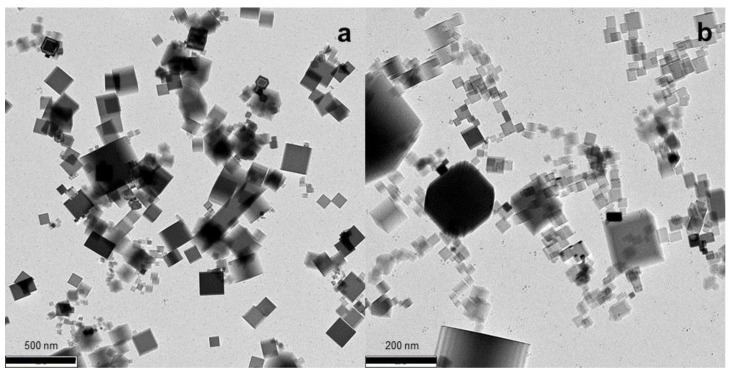
TEM images of MgO nanoparticles collected from the flame (**a**) without ions applied; (**b**) with positive ions applied.

**Figure 3 materials-14-02891-f003:**
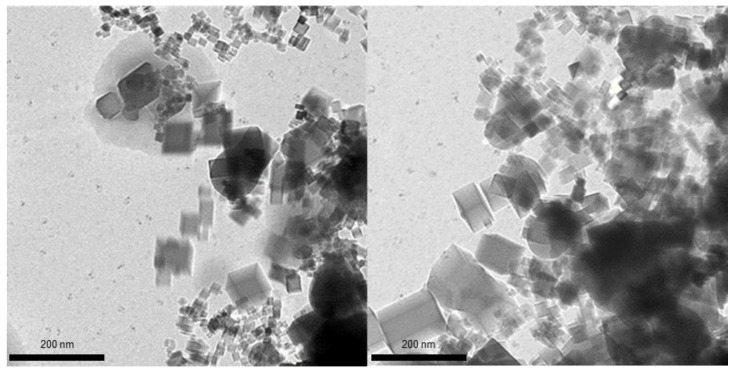
TEM images of MgO nanoparticles collected from flame with negative ions applied. Non-cubic particles and non-sharp particle edges are clearly seen.

**Figure 4 materials-14-02891-f004:**
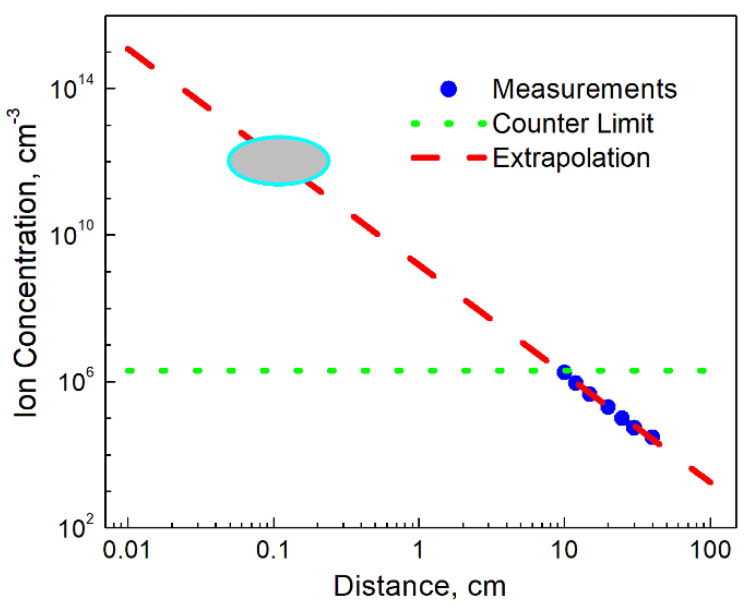
Results of ion concentration measurements. Experimental points, the counter limit and the extrapolation curve are presented. The ellipse of the graph shows the region, in which the ion concentration could achieve the level sufficient to affect the thermionic emission.

## Data Availability

The data presented in this study are available on request from the corresponding author.

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
