# Peer review of "Effect of External Charging on Nanoparticle Formation in a Flame"

_materials, 2021, doi:10.3390/ma14112891_

Round 1

Reviewer 1 Report

Since the main experimental results in this paper are TEM images in Figures 2 and 3, a clear interpretation of these images is critical. The authors wrote:

“The TEM images of nanoparticles collected from the corresponding flame are shown in Figure 3. As is clearly seen, the nanoparticles have lost their ideal cubic shape; the edges are no longer so sharp, and some of the nanoparticles tend to be spherical.”

Obviously, the second sentence here states that ALL the nanoparticles have lost their ideal cubic shape, and the edges of ALL OF THEM are no longer so sharp. However, it is clearly seen in Figure 3 that many particles have not lost their ideal cubic shape, and their edges remain as sharp as those of particles shown in Figures 2a and 2b. Further, the quality of the images in Figure 3 is poor, worse than the quality of images in Figures 2a and 2b. Why cannot the authors make better images?

In Figure 4, the dashed line is labeled as “interpolation,” while it is obviously extrapolation.

Author Response

Since the main experimental results in this paper are TEM images in Figures 2 and 3, a clear interpretation of these images is critical. The authors wrote:

“The TEM images of nanoparticles collected from the corresponding flame are shown in Figure 3. As is clearly seen, the nanoparticles have lost their ideal cubic shape; the edges are no longer so sharp, and some of the nanoparticles tend to be spherical.” Obviously, the second sentence here states that ALL the nanoparticles have lost their ideal cubic shape, and the edges of ALL OF THEM are no longer so sharp. However, it is clearly seen in Figure 3 that many particles have not lost their ideal cubic shape, and their edges remain as sharp as those of particles shown in Figures 2a and 2b. Further, the quality of the images in Figure 3 is poor, worse than the quality of images in Figures 2a and 2b. Why cannot the authors make better images?

We agree with the Reviewer’s statement in the need in a clear interpretation. To respond, we must admit that we did not make it clear by using the word “ALL”; we rather meant major alteration of the general scenario then comparison of each individual particle. To clarify, we revised the corresponding sentence as: “As is clearly seen, many nanoparticles have lost ideal cubic shape; their edges are no longer so sharp, and some of the nanoparticles tend to be spherical.”

With regards to the image quality, we produced many images, but unfortunately, the ones used in Fig 3 are of the best quality out of all of them. We must admit that currently we have no access to TEM of better quality. On the other hand, the reported effect is clearly seen in Figure 3 enabling the readership to follow our explanations.

In Figure 4, the dashed line is labeled as “interpolation,” while it is obviously extrapolation.

Thank you for catching this. The label is corrected.

Reviewer 2 Report

Authors made only minor corrections in comparison to the first version of the paper. In my opinon presented arguments are not sufficient to confirm the conclusions. Phenomenon discussed in the paper requires a deeper analysis.

Author Response

Authors made only minor corrections in comparison to the first version of the paper. In my opinon presented arguments are not sufficient to confirm the conclusions. Phenomenon discussed in the paper requires a deeper analysis.

As we stated in the previous rebuttal, the most recent advancements in the field have demonstrated that the temperature measurements, the Reviewer suggested, are currently impossible in the system of question due to demonstrated issues with interpretation. Such reality currently prevents more detail system characterization; however, we completely agree with the Reviewer that the phenomenon discussed in the paper will stimulate further research, which is one of the most important outcomes of this investigation.

Reviewer 3 Report

The reported data in this paper are interesting even if they are reported too succint to my opinion.

I have few remarks listed here:

What is the size of these nanoparticles? Are they polydisperse? What is the sigma g of their distribution?  

I don’t see exactly the goal of the ionizer. Is it to neutralize the particles because they are highly charged or to charge them because they are neutral?

Indeed particles from flame are self-charged (both polarities) even when the diameter is very small (< 10 nm):

Fang et al https://doi.org/10.1080/02786826.2017.1331028

Wang et al. : https://doi.org/10.1080/02786826.2017.1304635

More details on the home made ionizer design and products. To be sure that the ions and gases (ozone? NOx?)  produced by the ionizer are not contaminating the flame products. This point needs to be highlighted.

Author Response

The reported data in this paper are interesting even if they are reported too succinct to my opinion.

We appreciate the Reviewer’s interest.

I have few remarks listed here:

What is the size of these nanoparticles? Are they polydisperse? What is the sigma g of their distribution?  

As we stated in lines 103-104: the typical particle sizes ranged between 20 nm and 200 nm that is consistent with our previous work Ref. [8], in which the particle size distributions were studied  in detail in our paper Ref. [8]. Sigma_g of ~1.45 was reported there.

I don’t see exactly the goal of the ionizer. Is it to neutralize the particles because they are highly charged or to charge them because they are neutral?

As soon as particles are charged due to the thermionic emission of electrons, varying the charge density in the particle vicinity should control the rate of that thermionic emission. Then, the goal of the ionizer is to affect the thermionic emission by generating the excess charge density around particles. The negative ions that have the same charge of electrons should reduce the thermionic emission rate, while the positive ions act otherwise.

Indeed particles from flame are self-charged (both polarities) even when the diameter is very small (< 10 nm):

Fang et al https://doi.org/10.1080/02786826.2017.1331028

Wang et al. : https://doi.org/10.1080/02786826.2017.1304635

Thank you for letting us know about this work, It would definitely help to study the charge distribution in our system. Unfortunately, due to a limited process time (not exceeding 10 s), the described technique is not directly applicable. We now refer to the above papers and added the following statement: “A detailed study of the charge distribution would be helpful, however, due to a limited process time, the existing technique such as that described in Ref. [21, 22] should be substantially modified.”

More details on the home made ionizer design and products. To be sure that the ions and gases (ozone? NOx?)  produced by the ionizer are not contaminating the flame products. This point needs to be highlighted.

Thank you for highlighting this. Yes, we tested the flow downstream the ionizer using SMPS to confirm that no particulate is generated, so the ionizer does not contaminate the flame products. The following statement was added:The ionizer output was tested by the Scanning Mobility Particle Sizer Spectrometer (DMA Model: 3080N, and CPC Model: 3775, TSI Inc., Shoreview, MN USA)) to ensure that no particulate was generated, so the flame products would not be contaminated.”

Round 2

Reviewer 1 Report

OK